# Derivation of Genetically Defined Murine Hepatoblastoma Cell Lines with Angiogenic Potential

**DOI:** 10.3390/cancers17183002

**Published:** 2025-09-14

**Authors:** Keyao Chen, Ahmet Toksoz, Colin Henchy, Jessica Knapp, Jie Lu, Sarangarajan Ranganathan, Huabo Wang, Edward V. Prochownik

**Affiliations:** 1Division of Hematology/Oncology, UPMC Children’s Hospital of Pittsburgh, Pittsburgh, PA 15224, USA; keyaoc24@outlook.com (K.C.); toksozac@pitt.edu (A.T.); cmh259@pitt.edu (C.H.); jhk51@pitt.edu (J.K.); jie.lu@chp.edu (J.L.); 2School of Medicine, Tsinghua Medicine, Tsinghua University, Beijing 100084, China; 3Kanuni Sultan Suleiman Training and Research Hospital, Istanbul 34303, Turkey; 4The University of Pittsburgh, Pittsburgh, PA 15213, USA; 5Department of Pathology, Cincinnati Children’s Hospital Medical Center, Cincinnati, OH 45229, USA; sarangarajan.ranganathan@cchmc.org; 6The Department of Microbiology and Molecular Genetics, The University of Pittsburgh Medical Center, Pittsburgh, PA 15213, USA; 7The UPMC Hillman Cancer Center, Pittsburgh, PA 15232, USA; 8The University of Pittsburgh Liver Research Center, Pittsburgh, PA 15213, USA

**Keywords:** β-catenin, hepatocellular carcinoma, Hippo, NRF2, p16^INK4A^, p19^ARF^, Wnt, YAP

## Abstract

Hepatoblastoma (HB), the most common childhood liver cancer, often shows aberrant expression and/or mutations of the β-catenin (B), YAP (Y) and/or NRF2 (N) transcription factors and can be modeled in mouse livers by overexpressing combinations of these (e.g., BY, BN, YN and BYN). Because very few human or murine HB cell lines exist, genetically defined murine HB cell lines would be a significant advance. We describe here the generation of cell lines from primary BN and YN tumors. Unexpectedly, one BN line shows a remarkable ability to trans-differentiate into endothelial cells and can accelerate in vivo tumorigenesis and angiogenesis. These and previously derived BY and BYN lines have similar sensitivities to four drugs used to treat HB. Our generic approach for cell line derivation should allow for the generation of additional lines driven by less common drivers. A collection of such genetically defined and well-characterized cell lines will facilitate studies that are impractical to perform in vivo.

## 1. Introduction

Hepatoblastoma (HB), the most common pediatric liver cancer, arises almost exclusively in children less than 4 years of age and is classified into several distinct histopathologic subtypes [1,2]. Despite this complexity, it harbors fewer recurrent genetic alterations than any other known childhood or adult malignancy [3,4]. In this regard, >70% of HBs carry missense or in-frame deletion mutations in the *CTNNB1* gene, which encodes the β-catenin transcription factor (TF), and ~50% carry missense mutations in or amplify the *NRF2* gene, which encodes the NRF2 TF [5,6,7,8,9,10,11,12]. These changes dysregulate the cytoplasmic-to-nuclear shuttling of the TFs between the cytoplasm and nucleus and result in their constitutive nuclear confinement and target gene deregulation. Under these circumstances, β-catenin’s regulation by Wnt growth factor signaling is lost, and NRF2 loses its normal responsiveness to and regulation by oxidative and xenobiotic stresses [9,10,11]. A third TF, YAP, is also commonly deregulated in HB [9,13,14,15]. Along with its close paralog and co-activator, TAZ, YAP is the terminal TF of the Hippo pathway that regulates organ growth and cellular proliferation in response to signals as diverse as cell–cell contact, thrombin and glucagon [16,17,18]. Although recurrent mutations have not been identified in the Hippo pathway in HB, as many as 70% of tumors show strong nuclear localization of YAP [9,13,14,15]. Finally, up to 50% of human HBs show evidence for hypermethylation and/or down-regulation of the *CDKN2A* gene, which encodes the tumor suppressors (TSs) p16^INK4A^ and p14^ARF^ (p19^ARF^ in mice) in partially overlapping reading frames [19,20,21,22,23,24,25,26].

Mouse models of HB have demonstrated that patient-derived and nuclear-localized mutations of β-catenin, such as the 90 amino acid in-frame deletion Δ90 (hereafter B) and the NRF2 missense mutation L30P (N), as well as a nuclear-localized S127A missense mutation of YAP (Y), can induce HBs when expressed in the liver in any pair-wise combination (BY, NY and BN) and that the triple combination (BYN) generates particularly aggressive tumors [10,12,15,27]. Each of these tumor groups, which can be induced via the hydrodynamic tail vein injection (HDTVI) of Sleeping Beauty (SB) vectors encoding the above-mentioned mutant TFs, demonstrates distinct features. For example, BY tumors are associated with a median survival of ~90 days and histologically resemble the most common “crowded fetal” HB subtype [10,12,15,27]. BN and YN combinations generate tumors associated with a 2-fold longer median survival, with the former being more differentiated and the latter more resembling undifferentiated hepatocellular carcinoma (HCC) with HB-like features [11]. Finally, BYN tumors, with a median survival of <30 days, also display the crowded fetal pattern but possess innumerable and highly characteristic fluid-filled cysts that often abut well-demarcated foci of necrosis [11,26].

Few established human HB cell lines currently exist, and even fewer are readily available; moreover, some, such as HepG2, originate from atypical patients and/or possess unusual molecular features that are not representative of the vast majority of human tumors [28,29]. In addition, human cell lines, regardless of how easily they could be generated, would be unlikely to express the precise combinations of oncogenic drivers that would be of interest to many investigators. Finally, in vivo studies aimed at characterizing tumor cell-immune system interactions are not feasible since human cell lines must be propagated in mice that are severely immunocompromised. Similarly, murine cell lines established from *MYC* oncogene-driven tumors more closely resemble the moderately differentiated HCCs with HB-like features originally described in mice bearing tumors driven by a doxycycline-inducible human *MYC* transgene [30,31,32]. More importantly, these MYC-driven tumors and cell lines contain none of the common HB-associated mutations described above. Additionally, an intact *Myc* gene is not necessary for HB initiation in mice since its genetic ablation in hepatocytes still allows for highly efficient BY-generated tumorigenesis [10,27]. These findings underscore the need for HB cell lines that are molecularly defined, are driven by clinically relevant oncogene and/or TS combinations and that represent different histopathologic variants. The availability of such cell lines, which could be reliably generated from murine HBs, would make them useful for in vitro genetic manipulation and/or studies requiring controlled and/or rapid changes in the extracellular environment. They would also provide ideal reagents with which to screen for new chemotherapeutic drugs and to determine how their susceptibility to these or more traditional agents is impacted, if at all, by the underlying molecular drivers. Finally, these cell lines could potentially be propagated as orthotopic, subcutaneous and/or metastatic tumors in the immunocompetent mice from which they originate, thus permitting studies of host-tumor interactions that are otherwise impossible to conduct with human cell lines [26].

Having failed to generate immortalized cell lines from any of the above B-, Y- and/or N-driven HB types in over 30 prior attempts, we recently succeeded in efficiently deriving eight BY and three BYN murine HB cell lines [26]. This was made feasible by using in vivo Crispr/Cas9-mediated gene editing to mutate the *Cdkn2a* TS locus at the time of tumor initiation [26]. These cell lines, as well as primary tumors, could be markedly growth-inhibited when wild-type (WT) p16^INK4A^ or p19^ARF^ expression was restored, thus attesting to their importance in tumorigenesis [26,33]. Ten of the above eleven cell lines were tumorigenic and retained the appearance of HBs when propagated subcutaneously in the immune-competent FVB/N mouse strain from which they originated [26]. They also grew as “pseudo-metastatic” lung and lymphatic tumors when administered to recipient mice via standard tail vein injections and could be maintained in vitro as tumor spheroids on non-adherent surfaces, upon which they developed hypoxic interiors, with some cells acquiring the properties of endothelial cells (ECs) [26].

We now describe the derivation and characterization of five additional immortalized HB cell lines originating from the more slowly growing BN and YN primary tumor types [26]. As with BY and BYN cell lines, the establishment of BN and YN cell lines also required concurrent *Cdkn2a* inactivation, which is a common feature of HBs [19,20,21,22,23,24,25,26]. Four of the cell lines could also be propagated as subcutaneous and metastatic tumors. Like BY and BYN cell lines, BN and BY cells expressed EC markers and assumed EC-like behaviors to variable degrees, either when allowed to form spheroids or when propagated as monolayer cultures under hypoxic conditions. When combined with BY and BYN cell lines, BN and YN cell lines could also be used to assess their relative sensitivities to four of the chemotherapeutic drugs most commonly used to treat HB. While not of human origin, these novel cell lines provide new opportunities to examine the differential behaviors and properties of molecularly defined HB subtypes. These include studies aimed at understanding the ability of tumor cells to trans-differentiate into ECs, how the immune system impacts tumor growth *c* and how different oncogene combinations and/or mutations alter responses to various new chemotherapeutic or biologic agents. Finally, they outline a generic approach to generating HB cell lines driven by less common combinations of driver oncogenes and tumor suppressors.

## 2. Materials and Methods

### 2.1. Animal Care and Husbandry

FVB/N and nu/nu mice (Jackson Labs, Inc., Bar Harbor, ME, USA) were housed in micro-isolator cages in a pathogen-free facility at UPMC Children’s Hospital of Pittsburgh and provided ad libitum with standard animal chow and water. All care, diets and procedures were performed in accordance with the Public Health Service Policy on Humane Care and Use of Laboratory Animal Research Guide for Care and Use of Laboratory Animals. All experimental details have been previously described and were approved by the University of Pittsburgh’s Institutional Animal Care and Use Committee (IACUC) [26]. Animal experiments were performed with regard for ethical standards and strictly adhered to the guidelines set forth by IACUC and the Guide for the Care and Use of Laboratory Animals.

### 2.2. Plasmids, Plasmid DNA Purification, Hydrodynamic Tail Vein Injections (HDTVIs) and Transfections

Sleeping Beauty (SB) vectors encoding a patient-derived 90 amino acid in-frame deletion (Δ90) of β-catenin (B), a patient-derived missense mutation (L30P) of NRF2 (N) and a missense mutation (S127A) of the Hippo pathway terminal TF YAP (Y) have been previously described and their pair-wise and triple oncogenic potentials have been demonstrated [11,12,15,27]. HDTVI inocula were administered to 4–6 wk old FVB/N mice over ~5 sec. in 2 mL of 0.9% sodium chloride. These contained 10 μg each of B + N or Y + N SB vectors, 2 μg of a non-SB vector encoding SB transpose and 2 μg each of 2 pDG458 Crispr/Cas9 vectors [27] each of which encoded 2 gRNAs (C1 + C2 and C3 + C4) directed against exon 2 of the murine Cdkn2a gene [26]. For in vitro transfection of HB cell lines, pSBbi-RP SB vectors encoding dTomato and puromycin N-acetyl transferase (Addgene, Inc., Watertown, MA, USA), along with wild-type (WT) or relevant mutant forms of p16^INK4A^ and p19^ARF^ derived from BY and BYN cell lines (2 μg each, plus 0.5 μg of the SB transposase) [27] were stably introduced into the indicated tumor cell lines using the Lipofectamine-3000 according to the directions of the vendor (Thermo Fisher, Inc., Pittsburgh, PA, USA).

To readily monitor the acquisition of endothelial cell (EC)-like properties, some cell lines were transfected in vitro with 2 μg of a previously described vector encoding EGFP under the control of the EC-specific Tie2/Tek promoter (Tie2-EGFP) [34]. Stable clones were selected in Geneticin (G418, Thermo Fisher, Inc., Pittsburgh, PA, USA) (400 μg/mL) and then pooled for all subsequent studies.

### 2.3. Establishment of Immortalized BN and YN Cell Lines

Immortalized cell lines were derived as described previously for BY and BYN cell lines [26]. Briefly, upon reaching maximum allowable size, BN and YN tumors (each harboring multiple *Cdkn2a* mutations) were excised, minced into 1–2 mm pieces, and washed 3 times in PBS. After digesting for 30 min in 0.1% trypsin at 37 °C (Sigma-Aldrich, Inc. St Louis, MO, USA), the tissue fragments were further dispersed by vortexing and vigorous pipetting and were then incubated in 4–5 100 mm tissue culture plates containing Dulbecco’s modified D-MEM + 10% FBS that was supplemented with L-glutamine and penicillin G/streptomycin (Corning Life Sciences, Corning, NY, USA). Over the next several days to weeks, tumor cells gradually attached to the surface of the plates but did not replicate. Subsequent to this, cells were trypsinized weekly, with those remaining tightly adherent to the plates being discarded. Eventually, small colonies of non-contact-inhibited cells appeared and were pooled and expanded further.

Cell line transfections were performed with purified plasmid DNAs using Lipofectamine-3000 according to the directions of the vendor (Thermo Fisher, Inc., Pittsburgh, PA, USA). Control cell lines were established with the appropriate “empty” vectors.

### 2.4. Growth Curves

2000 viable tumor cells were seeded into individual wells of 12-well plates and maintained under standard growth conditions as previously described [26]. Counts were performed at regular intervals on 3–4 replicas using an Incucyte SX5 live cell imaging instrument (Sartorius Instruments, Göttingen, Germany).

### 2.5. Tumorigenicity of Immortalized YN and BN Cell Lines

YN or BN cell monolayers were trypsinized, washed twice in PBS and re-suspended in PBS at a concentration of 5 × 10^6^ cells/mL. A total of 0.2 mL was delivered via subcutaneous injection into the flanks of 4–6 week-old FVB/N or nu/nu mice. To determine the “pseudo-metastatic” potential of the cell lines, the same number of cells was delivered via slow tail vein injection. Mice were observed twice weekly for the appearance of palpable tumors. Tail vein-injected mice were sacrificed after 6–8 weeks or before if noted to be in any distress.

### 2.6. Tumor Histology and Confocal Microscopy

Subcutaneous and lung tumors were excised, immediately fixed in PBS-4% paraformaldehyde and then processed for H&E staining at the UPMC Children’s Hospital of Pittsburgh Core Histology Facility using standard procedures [12,26]. EGFP in tumor sections was examined following the embedding of fresh tumor tissue in O.C.T. compound (Thermo Fisher, Inc., Pittsburgh, PA, USA). After staining with Hoechst 33342, fluorescence imaging was performed with an Olympus Fluoview 1000 confocal microscope (Tokyo, Japan).

### 2.7. Tumor Cell Responses to Hypoxia

The effects of hypoxia on tumor cells were tested in 2 different ways. First, logarithmically growing tumor cells were trypsinized, re-suspended at a concentration of ~10^6^ cells/mL in DMEM-10% FBS and re-seeded into 15 mL polypropylene tubes (2 mL/tube). The tubes were incubated under normoxic conditions (21% O_2_ + 5% CO_2_) in a standard tissue culture incubator at a 30° angle with loosely attached caps that maintained sterility while allowing for gas exchange. Under these conditions, tumor spheroids formed rapidly and could be maintained in a highly viable state for at least 1–2 weeks. Small numbers of spheroids were removed periodically for staining with the hypoxia-sensitive dye Image-IT Green (Thermo Fisher, Inc., Pittsburgh, PA, USA) as previously described [26]. The same cells, stably transfected with the above-described Tie-2-EGFP vector [34], were also periodically examined by fluorescence microscopy for the expression of EGFP as a general surrogate for the expression of EC-specific genes. In other studies, the above cells were cultured as monolayers under standard normoxic conditions for 1–2 days and then exposed to an atmosphere of 1% oxygen + 5% CO_2_ in a Heracell Vios 160i CO_2_ incubator (Thermo Fisher, Inc., Pittsburgh, PA, USA) at 37 °C for the indicated periods of time, followed by image-IT Green staining or imaging for EGFP expression.

### 2.8. Isolation of EGFP+ Cells

Monolayer cultures of BN cell lines stably transfected with Tie2-EGFP in early log-phase growth were exposed to 1% oxygen for 48 h as described above. Cells expressing the highest levels of EGFP were then isolated by fluorescence-activated cell sorting using a Becton-Dickinson FACSAria III instrument (Franklin Lakes, NJ, USA) and then further expanded under normoxic conditions.

### 2.9. SDS-PAGE and Immunoblotting

Livers, tumors and cells were washed in PBS, snap-frozen in liquid nitrogen and stored at −80 °C until being further processed and analyzed as described previously [26]. Standard SDS-lysis buffer was prepared in the presence of protease and phosphatase inhibitors according to the directions of the supplier (Thermo Fisher, Inc., Pittsburgh, PA, USA) and as previously described [12,26,27]. Tissues were rapidly homogenized in a Bullet Blender (Stellar Scientific, Baltimore, MD, USA), diluted 1:1 in 2 x SDS-PAGE running buffer, boiled for 10 min and then clarified by centrifugation before storing at −80 °C in small aliquots. Immuno-blotting was performed using PVDF membranes and semi-dry transfer conditions [26,27]. Antibodies used included those directed against p16^INK4A^ (#Ab211542, 1:2000, Abcam, Cambridge, UK), p19^ARF^ (#NB200-174, 1:1000, Novus Biologicals, Centennial, CO), GAPDH (#G8795, 1:10,000, Sigma-Aldrich, Inc. St Louis, MO, USA), YAP (#4912, 1:1000, Cell Signaling Technologies [CST], Inc., Danvers, MA, USA). Additional antibodies included horseradish-peroxidase (HR)-conjugated goat anti-mouse IgG (#7076, 1:10,000, CST), HRP-goat anti-rabbit IgG (#7074, 1:5000, CST), and HRP-goat anti-rat IgG (#7077, 1:5000, CST). All immuno-blots were developed using a Pierce Enhanced ECL Chemiluminescence Detection Kit according to the directions of the vendor (Thermo Fisher, Inc., Pittsburgh, PA, USA). Signal relative to GAPDH was quantified by density scanning using ImageJ (1.54k) software.

### 2.10. Cdkn2a Exon 2 Amplicon Sequencing

Briefly, DNA was extracted from cell lines harboring *Cdkn2a* mutations using DNeasy columns according to the directions of the supplier (Qiagen, Inc., Germantown, MD, USA). Exon 2 of the *Cdkn2a* locus was amplified by PCR using different barcoded primers and the previously described conditions [26]. PCR products were combined and subjected to deep sequencing using the Illumina MiSeq platform (Azenta Life Sciences, Inc., Chelmsford, MA, USA). Sequencing reads were imported into CLC Genomics Workbench 24 (Qiagen), merged and de-multiplexed into individual groups that were determined by the identities of the barcoded primers. Individual reads from each group were then aligned to the mouse *Cdkn2a* locus.

### 2.11. RNAseq and Bioinformatics Studies

Total RNAs were isolated as previously described using RNeasy columns (Qiagen, Inc., Germantown, MD, USA). Only samples with RIN values >8.0 were processed further. Library preparation and sequencing were performed at the Pittsburgh Liver Research Center’s Core Sequencing laboratory (https://livercenter.pitt.edu/gsbc/#Sequencing) using an AVITI/Illumina instrument. Analyses were performed using nf-core/rnaseq version 3.12.0 as previously described [26]. Genes with low expression (average counts among all groups < 1) were filtered out. Normalized counts were generated by DESeq2 and then used for statistical quantification and heat map generation. Gene set enrichment analysis (GSEA) was performed using the clusterProfiler (4.16.0) R package. Endothelial gene sets for GSEA were achieved by filtering all “C8” gene sets from the Molecular Signatures Database (MSigDB) with “ENDOTHELIAL” in their names. Data from The Cancer Genome Atlas Program (TCGA) were accessed via UCSC Xena browser with batch-normalized TPM used for quantification. Only samples with both survival and RNAseq data were selected for analyses. Principal Component Analysis (PCA) and k-means clustering were performed to sub-classify tumors into different clusters. Survival curves were generated using the survival R package. *p*-values were calculated using the log-rank test and adjusted with the Benjamini–Hochberg (BH) method for multiple comparisons.

### 2.12. Drug Sensitivity Studies

Monolayer cultures were trypsinized and 3 × 10^3^ cells, in a volume of 100 μL, were seeded in D-MEM-FBS into individual wells of 96-well plates. The following day, fresh medium containing the indicated concentrations of drugs was added for 72 h, at which point standard MTT assays were performed as previously described (6 replicas/dose) [35,36,37]. Assays were read on a SpectraMAX Plus Micro Plate Reader (Molecular Devices, Inc., San Jose, CA, USA) spectrophotometer. Drugs used included cisplatin (Santa Cruz Biotechnology Inc., Dallas, TX, USA), etoposide (Sigma-Aldrich, Inc. St Louis, MO, USA), doxorubicin (Sigma-Aldrich, Inc. St Louis, MO, USA) and vincristine (Sigma-Aldrich, Inc. St Louis, MO, USA).

### 2.13. Statistical Analyses

R software v4.4.0 (R Foundation for Statistical Computing, Vienna, Austria) and GraphPad Prism v9.00 (Dotmatics, Inc., Boston, MA, USA) were used for statistical analyses. The ComplexHeatmap package was utilized for heatmap visualizations. The number of samples per group (n) for each experiment is indicated either in the figure legend or within the figure itself. Two-tailed, unpaired *t*-tests were used to compare significance between normally distributed populations and two-tailed Mann–Whitney exact tests were used to determine the significance between non-normally distributed populations. For comparisons among more than 3 groups, one-way ANOVA was performed to determine the variation among the groups, and multiple comparison was performed for pairwise comparisons of the mean value between selected groups.

## 3. Results

### 3.1. Efficient Generation of BN and YN Cell Lines

To derive immortalized cell lines from primary BN and YN HBs, we relied upon a previously used strategy that permitted the generation of 11 BY and BYN cell lines with 100% efficiency [26]. For this, Sleeping Beauty (SB) vectors encoding B + N or Y + N were delivered via HDTVI along with a non-SB vector encoding SB transposase [12,26]. While these two combinations alone generated tumors with >95% efficiency, key to the establishment of immortalized cell lines was the inclusion of non-SB-based Crispr/Cas9 vectors that allowed for mutagenesis of exon 2 of the *Cdkn2a* locus [26]. *Cdkn2a* encodes the p16^INK4A^ and p19^ARF^ (p14^ARF^ in humans) TS proteins in overlapping reading frames and is silenced in a significant fraction of human HBs [24,25,26,38]. Re-expressing either of these TSs in BY and BYN cell lines or primary tumors is also markedly growth-suppressive [26,33]. The gross appearances of the BN and YN tumors generated by this method were indistinguishable from those described previously in the absence of *Cdkn2a* targeting (Figure 1A) [11]. Examination of H&E-stained sections showed that all tumor histologies resembled those previously reported, thus indicating that *Cdkn2a* inactivation also did not significantly affect the microscopic appearance of tumors (Figure 1B) [11]. BN tumors also contained a somewhat more extensive vascular network (Figure 1B).

As reported for BY and BYN tumors harboring *Cdkn2a* mutations [26], two BN and three YN cell lines were readily established after 4–8 weeks of in vitro maintenance. Culturing these on coverslips in vitro showed that, with the exception of YN1 cells, which resembled previously described BY and BYN cells, BN and YN cells had a more epithelial appearance (Figure 1C) [26]. All five cell lines displayed similar doubling times (25–30 h), and the growth of four could be markedly suppressed by enforcing the expression of wild-type (WT) p16^INK4A^ or WT p19^ARF,^ with the latter being somewhat more potent (Figure 1D). The BN2 cell line could not be examined in this manner as it was resistant to transfection.

### 3.2. BN and YN Cell Lines Express Recurrent Truncated and/or Fused p16^INK4A^ and p19^ARF^ Variants That Are Distinct from Those Expressed by BY and BYN Cell Lines

Our previous deep sequencing of BY and BYN HBs revealed a large number of unique p16^INK4A^ and/or p19^ARF^ mutations that were initially generated by in vivo Crispr/Cas9 targeting of the *Cdkn2a* locus [26]. However, only a subset of these persisted in tumors and cell lines, and an even smaller subset was expressed in the form of p16^INK4A^ and p19^ARF^ truncations and/or fusions of varying complexities. Similar trends were observed in each of the five BN and YN cell lines, where we detected an average of 33 unique mutations (range = 29–38), with the top five comprising 38–79% of the entire mutational burden (Figure 2A and Appendix A). As with BY and BYN cell lines, immuno-blotting showed unique expression patterns for p16^INK4A^ and p19^ARF^ mutants in each cell line, with only a small number of all possible mutant proteins actually being detected. These ranged from only a single truncated form of p16^INK4A^ in YN1 cells in the absence of any p19^ARF^ to mutant or truncated forms of p19^ARF^ in BN1 cells in the absence of any p16^INK4A^ (Figure 2B).

More so than in BY and BYN cell lines, the *Cdkn2a-derived* mutant proteins expressed by BN and YN cells were often recurrent (Figure 2A) [26]. Such examples included the C130W and F152V point mutations seen in both the BN1 and BN2 cell lines, as well as the more complex BN2-5β, YN2-5β and YN3-3β mutations. Additionally, none of the mutations selected in BN and YN cell lines matched those previously generated in BY and BYN cell lines. This may have been a result of the reduced complexity of the Crispr/Cas9 vector mix used to target the *Cdkn2a* locus in the former tumors [26] and/or to the oncogenic backgrounds that were differentially permissive for the selection of mutant subsets.

To examine the latter possibility, we relied upon six previously described p16^INK4A^/p19^ARF^ mutants that were expressed at high levels in BY or BYN cell lines despite exerting varying degrees of growth suppression when overexpressed (Figure 2C) [26]. Comprised of p16^INK4A^ and p19^ARF^ truncations, fusions and internal deletions, all were V5 epitope-tagged to allow their expression levels to be directly compared (Figure 2C) (26). We transfected these individually into separate cultures of BN1 or YN2 cells and assessed their transient expression 2 days later in half the population, with the remaining half being selected in puromycin for 2 or 3 weeks (Figure 2D). A separate transfection of BY1 cells served as a control for the differential selection of each mutant [26]. In addition, a vector encoding WT p16^INK4A^ was used as a control for negative selection of the vectors, whereas an empty vector was used as a neutral selection control and was co-transfected with each of the other vectors (Figure 2D) [26,33]. Immunoblotting with antibodies against p16^INK4A^ and the V5 epitope allowed all proteins to be identified 2 days after transfection. As expected, WT p16^INK4A^ was initially expressed at similar levels in all three cell lines but was lost in the stably transfected populations upon further passage, whereas expression of each of the mutants was lost or retained in cell line-specific ways (Figure 2E). For example, the BY2-1α mutant was retained in all three cell lines, whereas the BY3-1α mutant was eventually lost in BY1 and YN2 cells but retained in BN1 cells.

The mutants BY2-2β and BY3-1β, as well as BY1-1β and BY2-1β, encoded proteins of similar sizes that could not be resolved. To identify which, if any, of these were selectively lost or retained in BN1 cells, we performed individual transfections with the empty pSBbi vector and one encoding each of the above four fusion proteins and selected these in puromycin for 21 days before repeating anti-V5 immunoblots. These results showed that BY3-1β and BY1-1β were selectively retained, whereas BY2-1β and BY2-2β were lost (Figure 2F). Finally, in separate studies, we transfected YN2 and BN1 cells with each of the above vectors and monitored subsequent proliferation in the presence of puromycin selection over the next 10 days. The differential growth suppression by these vectors closely agreed with the previous immuno-blotting experiments (Figure 2G). Collectively, these results emphasize that each cell line displayed a unique response to the suppressive effects of different p16^INK4A^/p19^ARF^ mutants and likely explain why different mutations among the cell lines were selected for in the first place.

### 3.3. Most YN and BN Cell Lines Retain Tumorigenic Potential

To determine whether BN and YN cell lines retain their oncogenic behaviors following in vitro establishment, we injected them subcutaneously (subQ) into the same FVB strain of immunocompetent mice from which they originated. Similarly to what we observed for 10 of the original 11 BY and BYN tumor cells [26], all 3 YN cell lines and one BN cell line were also tumorigenic. Histologic examination of these indicated that they retained the appearance of the originally described primary tumors, with BN1 tumors resembling differentiated HBs and YN1-3 tumors having both HB and variable degrees of HCC-like features (Figure 1B and ref. [11]). Unlike primary BN tumors, BN cell subQ tumors were not associated with a more prominent vasculature (Figure 3A and Appendix A). When the tumor cells were delivered to the lungs via slow tail vein injection, tumor nodules with histologies resembling those of the subQ and primary hepatic tumors could be obtained for two of the YN cell lines (Figure 3B and Appendix A). Thus, YN and BN tumors retain the histologies of the original HBs from which they originate, regardless of whether they are propagated as subcutaneous or “pseudo-metastatic” tumors.

### 3.4. BN and YN Cell Lines Form Spheroids and Acquire Endothelial Cell-like Properties in Response to Hypoxia

BY and BYN cells were previously shown to form anchorage-independent tumor spheroids that, in response to internal hypoxia, variably upregulated the expression of an EGFP reporter under the control of the endothelial cell (EC)-specific Tie2/Tek promoter [26,34]. We suggested that this might reflect attempts to re-establish normoxic interiors via the direct “trans-differentiation” of tumor cells into those expressing EC-like properties and potentially capable of generating a neovasculature [34,39,40,41,42]. Such EGFP+ cells were previously observed in subQ tumors formed by BY cells and were particularly prominent adjacent to the luminal endothelium of blood vessels [26].

To examine this in BN and YN cell lines, and to allow direct comparisons with our previous results, we first showed that all monolayer cell lines, regardless of their origin, responded similarly to staining with the hypoxia-detecting dye IT-Green after exposure to a 1% O_2_ environment (Figure 4A). However, when the same cell lines were allowed to form spheroids by culturing them on a non-adherent surface for 5 days under normoxic conditions, different degrees of IT-Green staining were observed, with the brightest spheroids being those formed by BYN and YN cells (Figure 4B). This indicated that, despite having similar appearances and sizes, the interiors of spheroids from each group varied considerably in their degree of hypoxia.

None of the above cell lines expressed Tie2-EGFP when maintained as monolayer cultures under normoxic conditions (Figure 4C). Hypoxia, however, induced strong EGFP expression in a subpopulation of BN cells but only faint, if any, EGFP expression in the other three cell types. The interiors of BN normoxia-maintained spheroids also showed strong EGFP expression, whereas those formed by the other three cell lines were again dimmer (Figure 4D). Collectively, these studies indicated that BN cells, more so than lines derived from the other 3 tumor types, were particularly prone to the induction of Tie2-EGFP.

To determine how long monolayer cultures of BN1-Tie2-EGFP cells retained EGFP expression, EGFP+ cells were isolated by fluorescence-activated cell sorting (FACS) following a 2-day exposure to 1% oxygen, expanded under normoxic conditions and evaluated periodically by fluorescence microscopy and flow cytometry. EGFP expression was retained for an extended period of time, with ~40% of cells remaining positive after one month and ~20% remaining positive after ~2 months (Figure 4E,F). EGFP+ cells isolated and expanded for ~2 weeks also formed EC-like “tubes” under hypoxic conditions, even in the absence of a basement membrane matrix that is typically needed to support the generation of these structures (Figure 4G) [34,43,44]. In contrast, unsorted BN1 cells did not form tubes under hypoxic conditions, even though a small number of EGFP+ cells again appeared. These observations, coupled with the fact that a relatively small fraction of Tie2-EGFP-transfected BN1 cells could be induced to express EGFP (Figure 4C), strongly suggested that only a small sub-population of tumor cells was capable of acquiring EC-like traits in response to hypoxic stress.

To determine whether any of the above-described Tie2-EGFP-transfected cell lines retained tumorigenic behaviors and could express EGFP in vivo, as we previously showed for BY1 cells [26], we generated subQ tumors and asked four questions regarding the biological behaviors of the EC-like BN1-Tie2-EGFP+ cells. First, could they affect the growth rates of subQ tumors if they were combined with BN1-Tie2-EGFP cells that had not been previously exposed to hypoxic conditions? Second, did a pure population of BN1-Tie2-EGFP+ cells remain tumorigenic? Third, did the presence of larger numbers of EGFP+ cells affect tumor histology and finally, how did the presence of large numbers of EGFP+ cells with EC-like properties affect tumor vasculature? To address these questions, we compared the properties of three groups of subQ tumors. The first group (control) was generated by injecting a pure population of BN1-Tie2-EGFP cells that had been grown under standard, normoxic conditions. The second group was composed of a 1:1 mix of BN1-Tie2-EGFP cells that had been maintained under normoxic or hypoxic conditions, with the latter group being mostly EGFP+. The third group was comprised exclusively of hypoxia-generated and sorted EGFP+ cells. In answer to our first question, each of the latter two groups of HBs grew significantly faster than the control group (Figure 4H). This finding also provided an answer for our second question, which showed that the third group of cells retained marked tumorigenic behavior despite their EC-like appearance. Although histologic examination of these three groups showed the tumor cells to be indistinguishable from one another (Figure 4I), the latter group contained a larger number of blood vessels, many of which were much larger than any of those in the control group (Figure 4I,J). Interestingly, examination of frozen sections from these tumors again failed to show any EGFP+ cells. Thus, the simplest interpretation of our findings is that EGFP+ EC-like cells contribute to early blood vessel formation in vivo but that EGFP expression and perhaps even other EC-like behaviors are transient and eventually disappear as they do in vitro (Figure 4F). Nonetheless, the presence of a pre-existing EC-like population provides significant growth and vasculogenic benefit to subQ tumor growth.

### 3.5. BN Tumors and Cell Lines Activate Unique Populations of EC-Specific Transcripts

Tie2-EGFP activation and in vitro tube formation by BN1 cells in response to hypoxia (Figure 4E–G) indicated that they likely express EC-specific genes other than *Tie2/Tek*. We, thus, examined the enrichment in these cells of 15 EC-specific gene sets retrieved from a variety of sources and found significant changes in each in at least one of the 4 murine tumor groups (Figure 5A). Moreover, the expression patterns of these gene sets relative to those of livers permitted each tumor group to be distinguished from the others. Similar results were found when only the 1853 non-redundant transcripts from the above 3750 members of these gene sets were examined (Figure 5B, Appendix A). A unique ~600-member subset of these genes, presumably originating from the liver sinusoidal ECs (LSECs) that normally comprise 15–20% of the organ’s cellular mass (Appendix A) [45,46], was largely downregulated in three of the four tumor groups, only to be replaced by numerous other non-LSEC-related sets of EC-specific transcripts from various members of the above-mentioned 15 gene sets. Interestingly, many LSEC-related genes continued to be expressed in BN tumors. These 1853 EC-specific genes were also overexpressed to variable degrees in subsets of 49 human HBs from two separate previously published studies (Figure 5C,D) [47,48]. Collectively, these findings indicate that, depending on their underlying genetic drivers, both murine and human HBs co-opt EC-specific genes from a variety of largely non-LSEC sources while also downregulating LSEC-related transcripts.

To examine the acquisition of EC-like gene expression profiles by tumor cells under more controlled conditions, we exposed unsorted, normoxic monolayer cultures of BN1-Tie2-EGFP cells (“UN”) to hypoxia (1% O_2_) for 2 days (“UH”), expanded the FACS-sorted EGFP+ population for ~10 days under normoxic conditions (“SN”) and then re-exposed the EGFP+ cells to 1% oxygen for 2 additional days (“SH”) (Figure 6A). Global gene expression profiling revealed 3646–5771 significant differences among the groups (Figure 6B), with many of these being due to anticipated changes in hypoxia-response genes in the UH and SH groups (Figure 6C). Endogenous Tie2/Tek up-regulation, while not appreciated in UH cells due to the relative paucity of those with EC-like properties (Figure 4C), was clearly evident in the sorted populations of SN and SH cells, thus supporting our previous use of EGFP induction as a surrogate marker for EC-like differentiation (Figure 6D).

Consistent with the idea that SN and SH cells originated from hepatocytes rather than an obscure population of ECs was the fact that they still expressed nearly 200 hepatocyte-specific genes, 10–15% of which were upregulated (Figure 6E). Notably, these included genes encoding α-fetoprotein, Prom1 and other proteins associated with more stem cell- or fetal-like hepatocyte precursors (Figure 6F) [49]. Finally, different subsets of the 1853 EC-specific genes described in Figure 5A were expressed among the 4 different sets of BN1 cells (Figure 6G).

To determine whether the dysregulation of the above 1853 EC-specific genes occurred in other tumors, we first examined their expression in highly undifferentiated murine HCCs that were generated in a rapid and reversible manner by the doxycycline-mediated induction of a human *MYC* transgene [30,32]. As observed previously in HBs (Figure 5B), ~600 of these transcripts were expressed in control livers and did not change significantly for the first 7 days after *MYC* induction, which was well before the appearance of tumors (Figure 6H). Large tumors that were sampled ~30 days after *MYC* induction showed a dramatic down-regulation of these transcripts and the up-regulation of a new set of previously non-expressed EC-specific transcripts. These largely disappeared 3 days and 7 days after *MYC* silencing, a time of marked tumor regression and tissue remodeling [30,32] and were replaced by a new set of ectopic EC-specific transcripts. The re-induction of HCCs 3–4 months after the initial tumors had completely regressed was associated with a pattern of EC-specific gene expression closely resembling that of the original tumors. From these studies, we conclude that three distinct patterns of EC-specific transcript expression are associated with the development and regression of *MYC*-driven HCCs. The first is associated with the previously described LSEC-specific transcripts of normal livers [47,48]. The second is associated with actively growing tumors and largely replaces the previous transcript group. Finally, there exists an additional unique set that is expressed only transiently during tumor regression.

We next examined the expression of the above EC-specific transcripts in various matched normal human tissues and tumors from The Cancer Genome Atlas. In the former samples, we determined that, as in the case of murine livers (Figure 5), specific subsets of the above EC-specific genes were expressed in highly tissue-specific ways, as expected from previous reports [45,46,50,51,52] (Figure 6I). In those cases for which a sufficient number of normal tissues could be evaluated, these expression patterns also differed significantly from those of tumors arising in adjacent regions, just as they had in murine HBs and HCCs (Figure 6J). Further refinement of these data from eight different tumor types showed that all could also be further subdivided into those with high, intermediate and low levels of expression of their specific transcripts (Figure 6K). In 6 of 8 cases, the subsets with the highest levels of EC-specific gene expression were associated with significantly shorter long-term survival (Figure 6L).

### 3.6. Y Suppresses EC Differentiation

Speculating that BN cells more readily acquire EC-like properties because Y suppresses EC differentiation [53,54], we co-transfected freshly sorted BN1-Tie2-EGFP+ cells (Figure 4E) with an empty (control) pSBbi SB vector or one expressing Y. Among the stable transfectants from the latter group, we found many fewer examples of EGFP + Y co-expressing cells as determined by fluorescence microscopy (Figure 7A). Quantification of the dTomato-positive cells by flow cytometry showed a ~20-fold reduced intensity of EGFP co-expression (Figure 7B). Additionally, sorted and expanded dTomato^high^ cells from this experiment expressed higher levels of exogenous Y than the unsorted population (Figure 7C). Consistent with previous reports that YAP negatively regulates both itself and the paralogous TF TAZ [55,56], endogenous Y levels were reduced in both populations of cells that expressed Y relative to those transfected with the control vector.

### 3.7. HB Cell Lines Have Similar Chemotherapeutic Drug Sensitivities

Current drug-based treatments for HB tend to be quite similar, even though a number of studies suggest that certain molecular signatures predict chemotherapeutic responses and long-term survival [2,5,31,57,58,59,60]. This approach differs from that taken with many other human cancers whose underlying oncogenic drivers are often used to inform therapeutic decisions and stratify patients [61,62,63,64,65,66]. Having multiple immortalized and molecularly well-defined HB cell lines provided an opportunity to examine this question empirically. We, therefore, compared the sensitivities of representative BY, BYN, BN and YN cell lines to four drugs that are commonly used to treat HB, i.e., cisplatin, etoposide, doxorubicin and vincristine. With few exceptions, all cell lines displayed similar sensitivities to each of these agents (Figure 8A,B). These studies, thus, indicate that the overall chemotherapeutic sensitivity of HB is little impacted by any of the four different driver oncogenic combinations and in the face of *Cdkn2a* locus inactivation.

## 4. Discussion

Histologic, biochemical and molecular differences among HBs have allowed for various classification systems that are intended to identify tumors with favorable or unfavorable survival [2,5,9,20,22,47,48,57,58,59,60]. However, the tumor’s relative rarity, the fact that it seldom appears after the age of 4 years and that its overall survival/cure rate is ~70% have made it challenging to conduct clinical studies aimed at individualizing chemotherapeutic regimens for these different groups while minimizing side effects, many of which are uniquely associated with this particularly young cohort [5,9,58,60,67,68].

The relatively small number of oncogenic drivers associated with HB classifies it as the least complex of human cancers [3,4,31,69,70]. Among the more common drivers, and associated with up to 80% of tumors, are missense and deletion mutations of *CTNNB1*, missense mutations or amplifications of *NRF2*, unspecified defects in the Hippo pathway that increase Y’s nuclear accumulation and silencing of the *CDKN2A* tumor suppressor locus, typically by promoter methylation [6,9,15,25,59,70,71]. Indeed, any pairwise or triple combination of B, Y and N overexpression is sufficient to promote HB initiation in mouse models of the disease, with concurrent *Cdkn2a* mutations being needed to derive immortalized cell lines from the primary tumors, as shown both here and elsewhere [6,12,15,47,48]. However, this superficial assessment belies what is almost certainly a more complex and nuanced aspect of the disease. For example, it seems likely that different combinations and levels of expression of the above factors are responsible for different tumor histologies and behaviors, that different oncogenic B mutations drive distinct patterns of target gene expression and phenotypes and that much of HB pathogenesis and behavior remains reliant on less common driver mutations and/or incompletely understood epigenetic changes [9,11,12,25,31,57,58,69]. This combinatorial complexity, along with the acknowledged paucity of and pressing need for human HB cell lines [26,29] underscores the appeal of cell lines derived from molecularly defined murine tumors.

In the current work, we derived and characterized immortalized cell lines from primary BN and YN HBs [11]. As with BY and BYN cell lines, this required the Crispr/Cas9-mediated mutational targeting and partial inactivation of p16^INK4A^ and p19^ARF^ and their respective downstream retinoblastoma and TP53 TS pathways [24,26]. We have previously shown that all such mutationally defined murine HBs and about half of human HBs, somewhat surprisingly, upregulate WT p16^INK4A^ and p19^ARF^ (Figure 2B) [26]. We have interpreted this as evidence that, despite the high-level expression of these two potent TSs, it remains insufficient to override the potent proliferative stimuli originating from the various overexpressed oncogenic drivers. The mutant forms of p16^INK4A^ and p19^ARF^ that are expressed as a result of in vivo Crispr/Cas9-mediated targeting of the *Cdkn2a* locus (Figure 2A,B) likely represent less growth suppressive versions of their wild-type counterparts that were selected from amongst the large number of other mutant forms of these proteins that were generated (Appendix A). While these various mutants remain capable of suppressing tumor cell growth, particularly when overexpressed (Figure 2E), they appear to be disabled enough to allow for immortalization and the establishment of permanent cell lines.

In addition to their tendency to recur, the most common p16^INK4A^ and p19^ARF^ mutations of BN and YN cell lines did not structurally resemble those of BY and BYN cell lines despite being generated in the same manner. This was supported by functional studies indicating that some mutants that suppressed BY and YN cell proliferation did not suppress BN cells (Figure 2E–G). The differential selection of p16^INK4A^ and p19^ARF^ mutants among the cell lines likely reflects the degree to which they influence and/or are influenced by the combinatorial signaling of each of the B, Y and N TFs that are expressed by each of the four tumor groups [9,11].

A major rate-limiting determinant of a tumor’s growth, and in some cases an indicator of its aggressiveness, is the degree to which it establishes an independent vasculature to supply oxygen and nutrients and dispose of waste products [72,73]. The means by which this is achieved include the classical recruitment of pre-existing blood vessels from adjacent normal tissues (neo-angiogenesis), the formation of a de facto neo-vasculature directly from tumor cells themselves (vasculogenic mimicry) and the direct trans-differentiation of tumor cells into those resembling ECs [34,40,41,42,74,75,76,77,78]. With regard to the latter, we have previously demonstrated that BY cells can acquire EC-like characteristics, but we have shown here that this process is even more impressive in BN cells (Figure 4). The acquisition of numerous EC-like properties is observed within 2 days following the exposure of BN monolayer cultures to hypoxia. Moreover, the subsequent formation of structures resembling EC-specific “tubes” appears to be quite robust as it occurs independently of the basement membrane substratum or growth factor supplementation that are typically needed to support this state (Figure 4G) [79,80]. These attributes, which persist for several weeks following this single, transient hypoxic episode, are accompanied by the expression of an array of EC-specific genes, many of which are shared between murine and human HBs (Figure 5A–D). In the former case, this is associated with expression patterns that clearly reflect the nature of the underlying molecular drivers (Figure 5A). This is in keeping with previous reports showing that EC-specific gene regulation is subject to control by the Wnt-β-catenin, Hippo and NRF2 pathways, whose terminal TFs not only include B, Y and N, respectively, but also share considerable positive and negative cross-talk [11,81,82,83]. The more variable EC-specific gene expression by human HBs likely reflects their greater overall molecular heterogeneity as previously described (Figure 5C,D) [47,48,69].

It is of interest that in no cases did the EC-specific gene expression signatures of different tumor groups or their cell lines resemble one another; nor did they precisely reflect those of normal livers, in which LSECs comprise 15–20% of the organ’s mass (Figure 5B–D and Figure 6G) [45,46]. However, the fact that the gene expression profiles of BN tumors and cell lines more closely reflect those of the liver than they do other HB cell types may explain why the former were predisposed to developing a more prominent vasculature (Figure 1B). The inability to do so nearly as well in locations other than the liver may reflect the restricted and aberrant nature of their EC-specific gene expression that does not allow for efficient trans-differentiation at other sites.

Importantly, the EC-like transcriptional profiles of our murine HBs and cells were not only distinct from one another but did not recapitulate any known EC subtype (Figure 5 and Figure 6G) [46,50,51,52]. This, coupled with the fact that the above-mentioned Wnt-β-catenin, Hippo and NRF2 cross-talking pathways are aberrantly regulated in these cells, suggests that the observed EC-specific gene expression profiles represent abnormal hypoxic responses in which genes from multiple EC subclasses are activated. As a result, only in some cases (e.g., in BN and less so in BY cells) [26] is the final collection of these transcripts sufficiently compatible to promote advantageous EC-like behaviors such as the ability to support tube formation or an actual vasculature (Figure 1A and Figure 4G).

Differences in EC-specific gene expression were also noted between multiple human cancer types and the normal tissues from which they originated (Figure 6I,J). As with their murine HB and HCC counterparts, none of the tumor-specific expression patterns resembled those of any of their respective normal tissues. Moreover, most tumor-specific expression patterns were also unique and could be further divided into subsets associated with significant survival differences (Figure 6K). This suggested that while many, if not all, cancers up-regulate subsets of EC-specific transcripts, only certain of these patterns confer a survival advantage. It remains to be determined whether these transcripts actually originate from tumor cells undergoing a trans-differentiation process such as we report here. Non-mutually exclusive alternative possibilities include the re-programming of ECs that comprise the initial tumor stroma or the recruitment of exogenous ECs from outside the tumor’s immediate vicinity. Irrespective of these source(s), our findings are consistent with others showing that tumors harboring large populations of ECs and/or having high microvascular density are often more aggressive and correlate with shorter survivals [84,85]. While hypoxia and its associated induction of HIF1α are clearly necessary for EC induction and function, the expression of functionally relevant genes likely requires additional factors, including the proper balance among other TFs such as B, Y and N (Figure 7) [86,87].

A potential explanation for why BN cells were particularly predisposed to EC-like trans-differentiation was suggested by the fact that they are the only ones that do not express Y. Indeed, the enforced overexpression of Y in BN cells markedly inhibited Tie2-driven EGFP expression and downregulated the expression of endogenous YAP (Figure 7C) [78,88]. In addition to this negative feedback control over its own promoter and that of its paralog, TAZ, YAP can either promote or inhibit EC differentiation in a context-dependent manner [13,53,54,55,56,74,78,82,88]. Interestingly, we did not find any evidence to indicate that human HBs with higher levels of EC-specific transcripts (Figure 5C,D) express less YAP or TAZ. Non-mutually exclusive reasons for this include the possibility that this regulatory pathway is not evolutionarily conserved, that human HBs are more molecularly heterogeneous or that EC-specific gene expression with actual functional consequences requires specific levels or types of β-catenin and/or NRF2 deregulation (i.e., different mutations or amplification), which can exert significant effects on the overall transcriptional profile of the tumor [12,13,27,47,48,69,89]. The trans-differentiation of ECs from other tumor types, such as gliomas and melanomas, is well-established and may explain their resistance to anti-angiogenic agents that target proteins such as vascular endothelial growth factor, which recruit pre-existing exogenous ECs residing in proximity to tumors [34,39,41,42,77,90,91].

A surprising finding from our studies was that BN1 cells exposed to hypoxia did not appear to retain their EC-like properties during in vivo tumorigenesis despite contributing extensively to growth and neo-vascularization (Figure 4H–J). Indeed, pure populations of EGFP+ BN1-Tie2-EGFP cells were able to generate HBs much more rapidly than control BN1 cells. This suggested not only that the EC phenotype was transient in vivo just as it is in vitro (Figure 4E,F) but that EC-like cells could revert back to tumor cells and generate tumors that were histologically indistinguishable from those formed by control BN1 cells (Figure 4I). This further suggested that the benefits conferred by EC-like BN1 cells likely occur early (i.e., prior to complete reversion back to tumor cells), perhaps by contributing to the early stages of tumor blood vessel formation and providing an environment that is more conducive to blood vessel formation by host-derived “true” ECs.

Although HB tends to respond well to chemotherapy and is associated with an overall long-term-survival/cure rate exceeding 70%, some tumors show poor responses to standard drug regimens or recur, in which case they are almost universally chemo-refractory [5,9,31,57,58,60,70]. Yet, despite certain pre-treatment molecular subtypes being associated with more unfavorable survival, current chemotherapeutic regimens do not account for these differences and, thus, do not stratify patients accordingly [7,57,59,60,69,70]. This is in marked contrast to many other cancers, where certain molecular subtypes are often assigned to different treatment cohorts [65,66,92,93,94]. We capitalized on the well-defined and distinct molecular properties of our four HB cell line groups [10,12,27] to ask whether they demonstrate any significant and consistent differences in their susceptibilities to chemotherapeutic drugs that are commonly employed to treat the disease. Although some differences were noted among individual cell lines within a specific molecular category, we were unable to make any significant inter-group distinctions (Figure 8). It is, thus, likely that factors other than B, Y and N are needed to impart the differential chemotherapeutic sensitivities that have been reported [5,57,58,69,70]. Important factors that are deserving of closer scrutiny include the degree to which the *CDKN2A* locus is inactivated, given that the Rb and TP53 pathways that are regulated by p16^INK4A^ and p19^ARF^ are major determinants of both survival and chemotherapy responses [23,24,25]. Nonetheless, the cell lines described here and elsewhere [26] should still be useful for the pre-clinical testing of new chemotherapeutic and/or biologic agents as they emerge.

Finally, despite the advantages of the above cell lines and those we have previously described [26], It is important to remain cognizant of their limitations. First, and foremost, they are not human and, thus, may display species-specific behaviors and properties that are not shared with their adult counterparts [95]. Second, and as is true for other murine cell lines, their driver oncogenes and not regulated in the same manner as they would be if they were under the control of their own, endogenous promoters. Third, the residual tumor suppressor activity of the mutant and fusion forms of p16^INK4A^ and p19^ARF^ (Figure 2), while essential for achieving immortalization, may not reflect the contribution, if any, that these genes make to establishing primary human HBs or HB cell lines [29]. In addition, they may alter downstream senescence and apoptosis pathways in ways that have little relevance to human tumors. Despite these shortcomings, our studies point out the way to address them while still providing valuable reagents. Ways of obtaining more relevant and human-based models may include such approaches as the use of mice with “humanized” livers and Crspr/Cas9-based targeting of endogenous human oncogene loci to generate mutations akin to those described here [96].

## 5. Conclusions

The work presented here and elsewhere [26] provides a means by which murine cell lines, with defined oncogenic drivers akin to those of human tumors, can be reproducibly isolated from primary HBs. In the vast majority of cases, they remain tumorigenic in immune-competent hosts, are capable of forming metastasis-like pulmonary lesions and retain the histologies of the tumors from which they originate. As such, they provide a much-needed set of well-defined reagents that have, heretofore, been lacking in the study of this disease [6,29]. We envision these cell lines as having future utility in the study of tumor metabolism, drug screening and the identification of previously unknown molecular susceptibilities. The acquisition of EC-like properties in response to hypoxia also provides a new model in which to more closely study the trans-differentiation process and the molecular events that are responsible for this reversible transition.

## Figures and Tables

**Figure 1 cancers-17-03002-f001:**
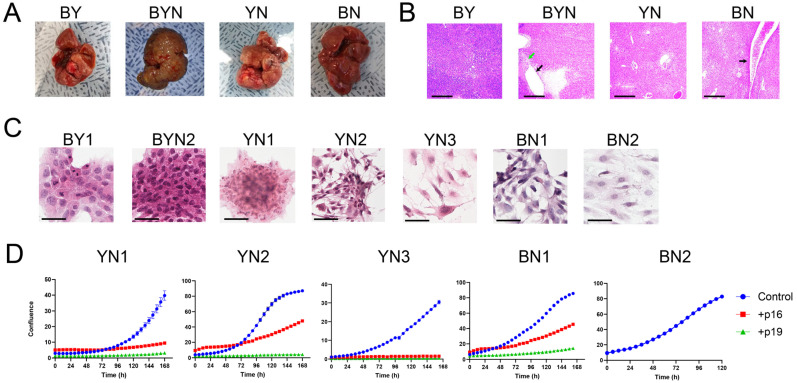
Properties of BN and YN HBs and their progeny immortalized cell lines. (**A**) Gross appearance of typical tumors generated by the indicated combinations of oncogenic drivers. Images of previously generated BY and BYN tumors [26] are included for comparison. To allow for the establishment of immortalized BN and YN cell lines, all tumors were generated with the inclusion of 2 Crispr/Cas9 vectors encoding 4 different gRNAs directed against exon 2 of the Cdkn2a locus [26]. (**B**) H&E-stained sections of tumors from (**A**). Note the more prominent blood vessels in BN tumors (black arrow) and the previously reported fluid-filled cysts adjacent to regions of necrosis in BYN tumors (black and green arrows, respectively) [11,26]. Scale bars = 500 μm. (**C**) H&E-stained cells from the indicated immortalized cell lines propagated on coverslips in vitro. BY1 and BYN2 cell lines were derived and characterized previously and are included here for comparative purposes [26]. Scale bars = 50 μm. (**D**) Growth curves of the indicated cell lines and their suppression via the enforced expression of WT p16^INK4A^ and p19^ARF^. The indicated cell lines were transfected with a control pSBbi-RP SB vector or with vectors encoding WT p16 ^INK4A^ or p19^ARF^. Two days later, the cells were seeded into 12-well plates and maintained in 2 μg/mL of puromycin while monitoring dTomato expression. Subsequent growth was monitored using an Incucyte S3 Live-Cell imaging and Analysis System. Each point represents the mean of 4 replicas +/− 1 S.E. Note that BN2 cells were resistant to transfection on multiple occasions.

**Figure 2 cancers-17-03002-f002:**
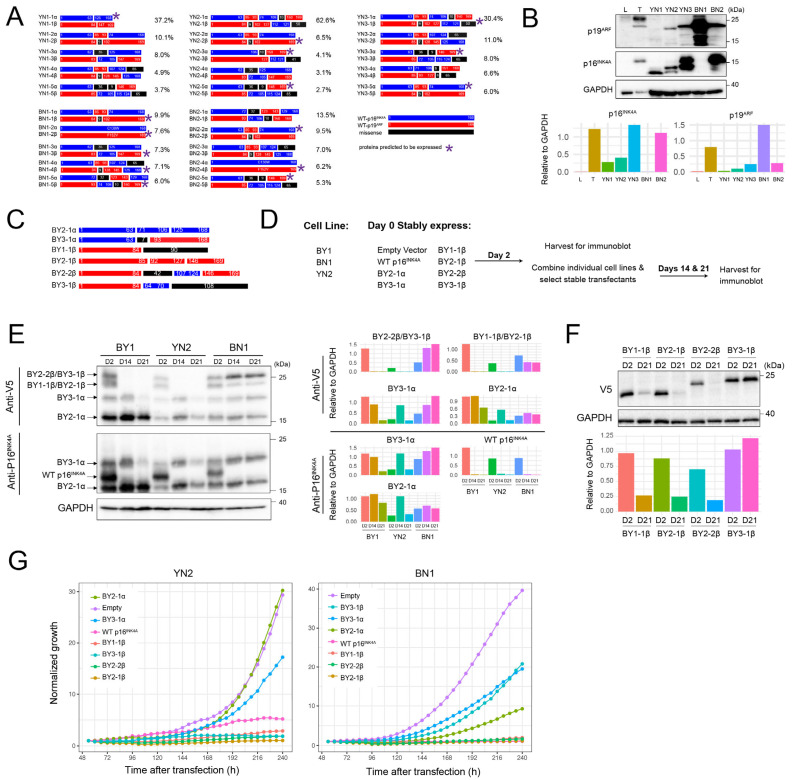
YN and BN cell lines express small, unique subsets of p16^INK4A^ and p19^ARF^ mutants. (**A**) Depictions of the proteins encoded by the 5 most abundant *Cdkn2a* mutations identified in each of the indicated cell lines. The frequencies with which they were detected based on deep sequencing of *Cdkn2a* exon 2 PCR products obtained from cell lines are indicated to the right of each cartoon. Asterisks indicate the mutants that were deemed the most likely to be expressed as proteins, based upon in silico translation of the open reading frames and the sizes of the actual proteins observed by immunoblotting (panel **B**). Appendix A contains a more comprehensive list of the mutations identified and their abundance. *: Proteins predicted to be expressed. (**B**) Immunoblots of mutant p16^INK4A^ and p19^ARF^ proteins expressed by the indicated cell lines. Included as controls were a sample of normal liver (L) and a primary BY HB (T) with intact *Cdkn2a* loci. The latter expressed WT p16^INK4A^ and p19^ARF^ as previously described [26]. Expression of each protein relative to that of GAPDH, as determined by densitometric scanning, is shown beneath the blot. (**C**) BY-derived mutant and/or fusion p16^INK4A^ and p19^ARF^ proteins were used for expression in BN and YN cell lines. All encoded proteins were V5 epitope-tagged to allow expression levels to be directly compared. See ref. [26] for previous characterization. (**D**) Approach to evaluating the growth suppressive effects of the mutants depicted in C on BN1 and YN2 cells. (**E**) Differential selection of p16^INK4A^/p19^ARF^ mutants. The indicated cell lines were individually co-transfected with pSBbi SB vectors encoding the mutant proteins depicted in panel (**C**) plus an equal amount of the empty pSBbi vector. Two days later, half the cells were used to assess the transient expression of each protein as described in (**D**). The remaining cells were selected in puromycin for 14 and 21 days and assessed for the expression of their respective protein at these times using anti-V5 or anti-p16^INK4A^ antibodies. Expression of each protein relative to that of GAPDH, as determined by densitometric scanning, is shown to the right of the blot. (**F**) Selective retention of the non-resolvable mutants shown in (**E**). BN1 cells were separately co-transfected with the empty pSBbi vector alone plus one encoding each of the four indicated mutants. Puromycin-resistant clones were then selected and expanded for 3 weeks as in (**E**), followed by immunoblotting to detect each of the V5-tagged mutants. Expression of each protein relative to that of GAPDH, as determined by densitometric scanning, is shown beneath the blot. (**G**). The indicated vectors were transfected into YN2 or BN1 cells, which were then seeded into 12-well plates 2 days later in the presence of puromycin and enumerated over the course of the next 10 days. Each point represents the mean of 4 replicas +/− 1 S.E. Original Western Blot images in Appendix A.

**Figure 3 cancers-17-03002-f003:**
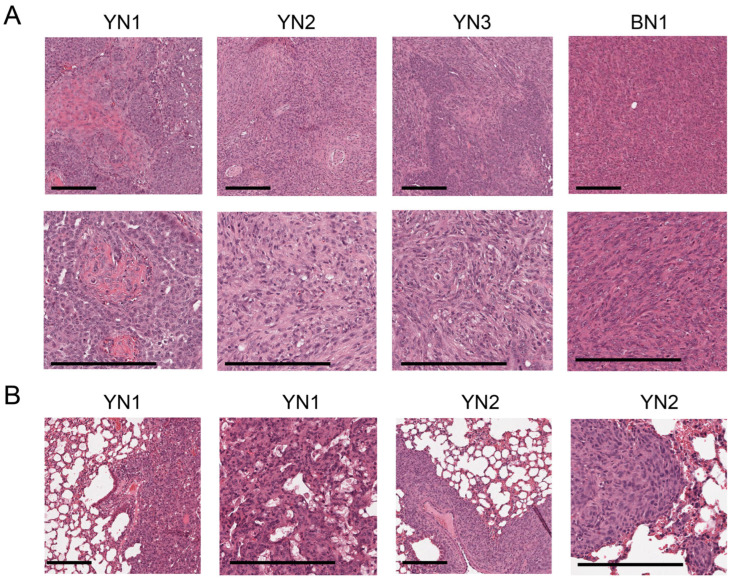
Histologies of BN and YN tumor cell lines following subQ or “pseudo-metastatic” growth in lungs. (**A**) H&E-stained sections of subQ tumors originating from the indicated cell lines. See Appendix A for additional examples. Scale bar: upper panel = 100 μm, lower panel = 200 μm. (**B**) H&E-stained sections of lung tumors generated by tail vein injection of the indicated cell lines. See Appendix A for additional examples. Scale bar: shorter = 100 μm, longer = 200 μm.

**Figure 4 cancers-17-03002-f004:**
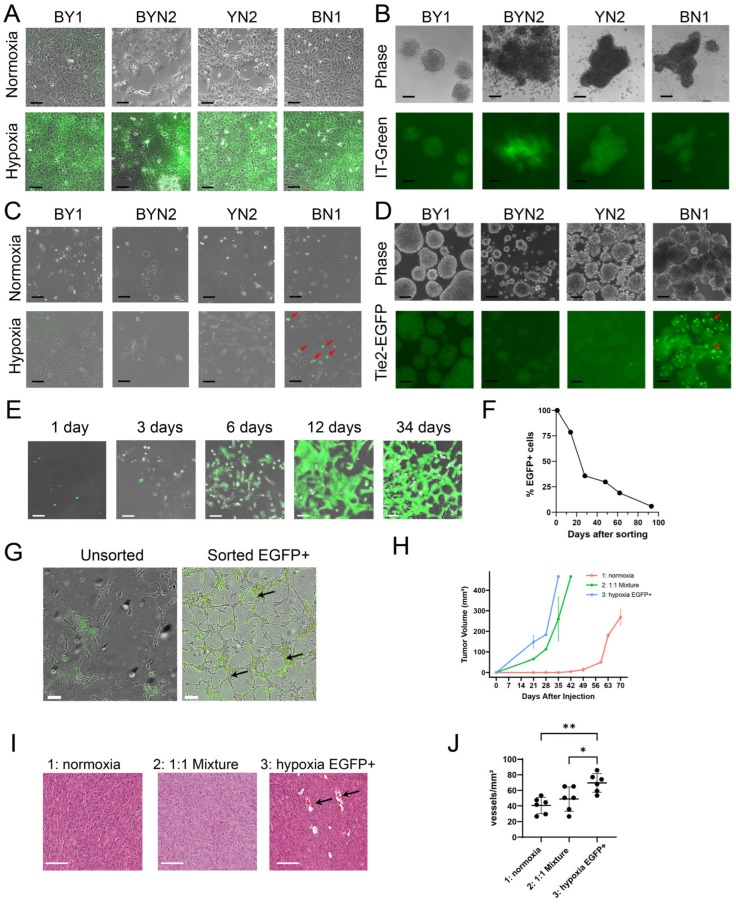
HB cell lines are variably susceptible to EC-like differentiation in response to hypoxia. (**A**) IT-Green staining of monolayer cultures of the indicated cell lines maintained under normoxic or hypoxic (1% O_2_ for 48 h) conditions. Scale bars = 100 μm. (**B**) IT-Green staining of tumor cell spheroids of the indicated types 5 days after formation. Scale bars = 100 μm. (**C**) Tie2-EGFP expression in the indicated monolayer cultures was maintained under normoxic or hypoxic conditions for 48 h. Arrows indicate EGFP+ cells that were particularly prominent among BN1 cells. Scale bars = 100 μm. (**D**) EGFP expression in the indicated stably transfected Tie2-EGFP cell lines after formation of spheroids for 5 days. Scale bars = 100 μm. (**E**) BN1-Tie2-EGFP cell monolayers were maintained in 1% oxygen for 2 days, at which point, EGFP+ cells were purified by FACS and then expanded and maintained under normoxic conditions for the indicated periods of time. Panels show merged phase contrast and fluorescence images. Scale bars = 100 μm. (**F**) After isolating pure EGFP+ cell populations from hypoxic BN1-Tie2 EGFP cells, they were maintained under normoxic conditions (**E**). At the times indicated, they were subjected to flow cytometry (ca. 10,000 cells/run) to quantify the percent of EGFP+ cells remaining. (**G**) BN1-Tie2-EGFP cells form EC-like tubes under hypoxic conditions. Left panel: Unsorted BN1-Tie2-EGFP cells were maintained continuously under hypoxic conditions for 6 days. Right panel: Sorted EGFP+ cells from (**E**) were expanded for ~2 weeks under normoxic conditions, re-plated and then maintained under hypoxic conditions for 6 days. Both panels show merged phase contrast and fluorescence images. Scale bars = 100 μm. Arrows indicate representative complete tubes that were formed by the latter group of cells. (**H**) BN1-Tie2-EGFP, maintained under normoxic conditions, were used to generate control subQ tumors in nu/nu mice (Group 1: red curve). A second group of mice was injected with a 1:1 mix of the same cells plus an EGFP+ population that was generated following exposure to 1% O_2_ for 2 days (Group 2: green curve). Finally, a third group of tumors was generated with a pure population of EGFP+ BN1-Tie2-EGFP cells (Group 3: blue curve). Tumor volumes were quantified weekly. In all 3 cases, a total of 10^6^ cells was injected. n = 4 mice/group. (**I**) Representative H&E-stained sections from the groups shown in (**H**), Scale bars = 100 μm. Black arrows indicate example of tumor micro-vasculature. (**J**) Quantification of blood vessel number from the tumors depicted in (**H**,**I**). 5–6 high-power fields from each group were randomly selected, and the blood vessel content of each was enumerated. *: *p* < 0.05; **: *p* < 0.01.

**Figure 5 cancers-17-03002-f005:**
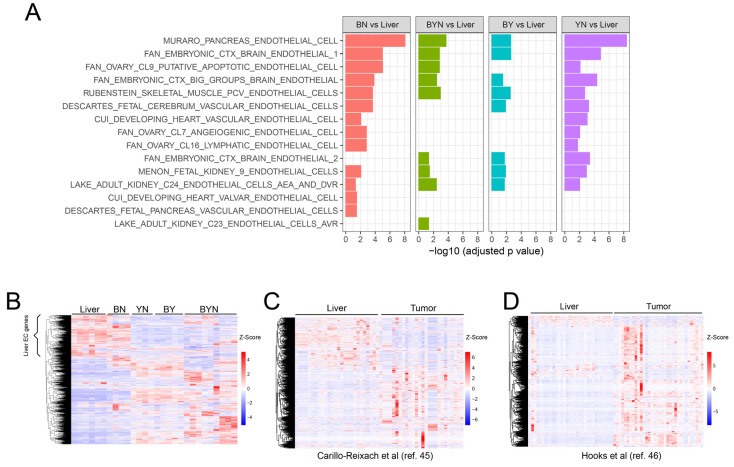
Murine and human HBs upregulate the expression of EC-specific transcripts. (**A**) Summary of GSEAs performed in primary murine HBs with 15 EC-specific gene collections from the MSigDB containing a total of 3750 transcripts. RNAseq results from our previously reported murine livers and HBs [11,12,27] were used to compare gene expression levels between each of the 4 possible tumor types relative to control normal livers. (**B**) Expression of 1853 unique transcripts extracted from the above 15 EC-specific gene sets in the normal murine livers and tumor groups from (**A**). The bracket indicates a subset of these transcripts that are likely expressed by LSECs. (**C**,**D**) Expression patterns of the human orthologs of the murine genes indicated by the brackets in A genes in 2 independent series of RNAseq studies performed on human HBs [47,48]. All scale bars represent the z-score.

**Figure 6 cancers-17-03002-f006:**
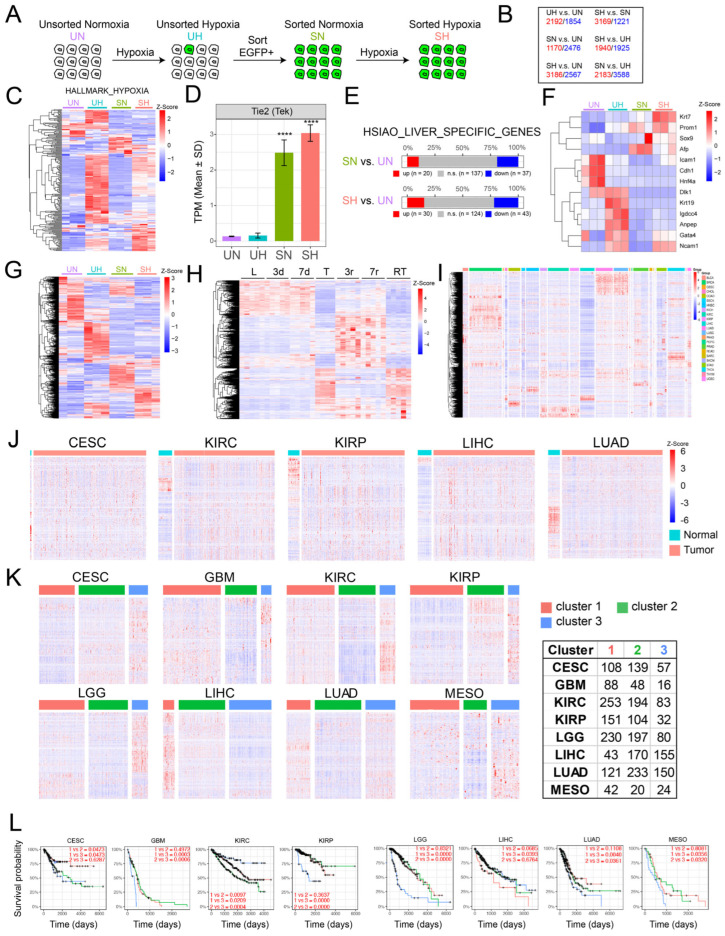
Gene expression profiling of BN1-Tie2-EGFP+ cells reveals widespread EC-specific gene expression within a more undifferentiated sub-population of hepatocytes. (**A**) Scheme for RNAseq profiling of BN1 cells maintained as monolayers under 4 sets of conditions: 1: unsorted cells maintained under normoxia (UN); 2: unsorted cells exposed to hypoxia (1% O_2_) for 2 days (UH); 3: FACS-sorted, EGFP+ cells expanded for ~10 days under normoxic conditions following a 2 day exposure to hypoxia (SN); 4: SN cells re-exposed to hypoxia for 2 additional days (SH). (**B**) Gene expression differences among BY1-Tie2-EGFP cells cultured under the 4 conditions described in (**A**). Quantification was performed on 3 replicas of cells from each of the 4 groups. Only genes showing q values < 0.05 and fold differences > 2 were included in the analyses. Red numbers: genes whose expression was higher in the groups shown at the left; blue numbers: genes whose expression was lower in the group shown at the left. (**C**) Heat map of gene expression differences among a group of 178 hypoxia-regulated genes obtained from the MSigDB (Appendix A). (**D**) Expression of Tie2/Tek in each of the 4 cell groups. (**E**) Expression of a group of 194 hepatocyte-specific genes among the indicated groups of cells (Appendix A). (**F**) Expression of 13 genes that are markers of immature hepatocytes and/or hepatocyte precursors [49]. (**G**) Selective expression among the indicated BN1-Tie2-EGFP cell lines of the 1853 non-redundant EC-specific genes derived from Figure 5B. For each of the indicated groups in (**C**–**G**), RNAseq was performed on 3 independent replicas. See Appendix A for a list of these genes, relative changes in expression and q values. (**H**) Differential expression of 1853 EC-specific genes from Figure 5B during the course of HCC generation in response to human MYC transgene induction [30,32]. L = control liver prior to *MYC* induction; 3d and 7d = days after MYC induction and prior to the appearance of any visible tumors; T = HCC after 30 d of MYC induction; 3r and 7r = 3 d and 7 d after MYC inactivation and the initiation of HCC regression; RT = recurrent HCCs induced ~3–4 months after the initial tumors had regressed. (**I**) Expression of human orthologs of the 1853 unique EC-specific genes from Figure 5 in 23 cancers and their matched normal tissues from TCGA. (**J**) EC-specific gene expression from selected normal tissues depicted in (**I**) along with tumors arising in adjacent regions. (**K**) EC-specific gene expression varies among different subsets of certain human cancers. Principal Component Analysis (PCA) followed by k-means clustering was used to sub-classify each of the indicated types of cancers based on differential levels of expression of the above 1853 EC-specific genes. The list to the right of the heat maps indicates the number of tumors included in each of the indicated categories. (**L**) Kaplan–Meier curves showing the survival of patients among different subgroups in (**K**). Adjusted *p*-values are displayed for pairwise comparison between different subgroups with Benjamini–Hochberg correction.

**Figure 7 cancers-17-03002-f007:**
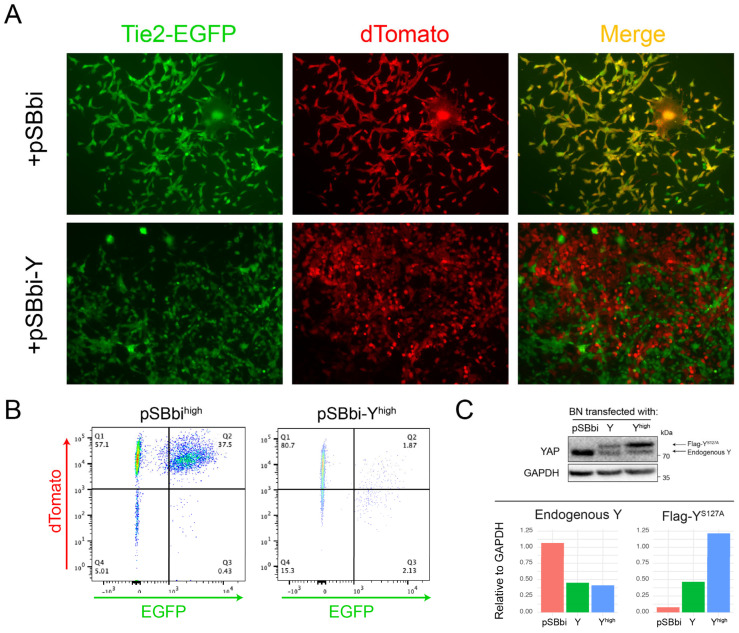
Exogenous Y overexpression suppresses Tie2-EGFP. (**A**) Following 2 days of hypoxia, BN1-Tie2-EGFP cells were sorted, expanded for 3–4 days and then transfected with a pSBbi-Y expression vector or the empty pSBbi vector as a control. Puromycin-resistant dTomato+ cells were then selected for 10 days and imaged by fluorescence microscopy. Merged images show that Y suppressed EGFP expression. (**B**) dTomato and EGFP expressions in the cells from (**A**) were quantified by flow cytometry. (**C**) dTomato^high^ and unsorted BN1-Tie2-EGFP cells from (**B**) were expanded and examined by immunoblotting using an anti-Y antibody. Endogenous and exogenous Y were distinguished by virtue of the latter bearing a FLAG epitope and migrating more slowly. The blot was scanned and quantified. The levels of endogenous and exogenous (Flag-tagged) Y, normalized to GAPDH, are indicated in the bottom portion of the panel. Original Western Blot images in Appendix A.

**Figure 8 cancers-17-03002-f008:**
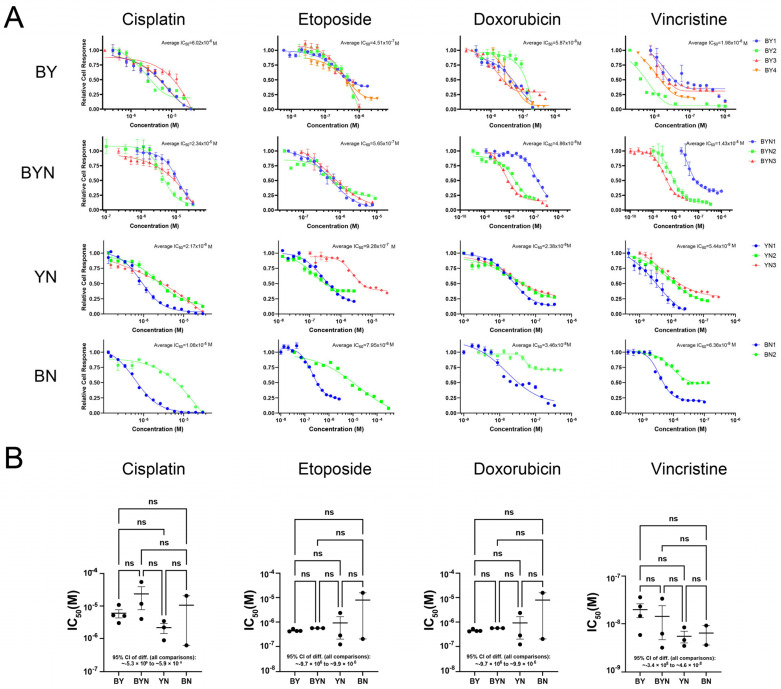
Murine HB cell lines display similar sensitivities to commonly employed therapeutic agents. (**A**) Typical dose–response curves. BY, BYN, BN and YN cell lines were seeded into 96-well plates, allowed to achieve log-phase growth over the next 24 h and then exposed to the indicated concentrations of drugs for 72 h. MTT assays were then performed, with each point representing the mean of 6 replicas +/− 1 SE. (**B**) Chemotherapeutic drug sensitivities of all HB cell lines tested. Each point represents the calculated IC50 for one individual cell line from the indicated molecular group based on dose–response profiles generated as described in (**A**). One-way ANOVA followed by multiple comparisons was performed to show no significant differences among different groups for each drug (with 95% CI reported in the figure). Error bars: the mean +/− SE within each group. ns: not significant.

## Data Availability

RNAseq data have been deposited in the NCBI Gene Expression Omnibus (GEO) under accession number GSE302803 and can be accessed at: https://www.ncbi.nlm.nih.gov/geo/query/acc.cgi?acc=GSE302803. Some of the results reported here relied upon data generated by the TCGA Research Network: https://www.cancer.gov/tcga (accessed on 4 May 2025). All other data supporting the findings of this study are available from the corresponding author upon reasonable request.

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
