# Peer review of "Derivation of Genetically Defined Murine Hepatoblastoma Cell Lines with Angiogenic Potential"

_cancers, 2025, doi:10.3390/cancers17183002_

Round 1

Reviewer 1 Report

Comments and Suggestions for Authors

Keyao et al. presents a comprehensive and well-structured study describing the generation of hepatoblastoma (HB) cell lines derived from BN and YN tumors, effectively addressing a critical gap in HB research. The rationale of the study is clearly articulated, and the authors convincingly demonstrate that CRISPR-mediated inactivation of Cdkn2a enables efficient derivation of immortalized HB cell lines. A particularly novel finding is the discovery of angiogenic plasticity in BN cells under hypoxic conditions. While the study is innovative and methodologically sound, several major concerns should be addressed before the manuscript can be considered for publication.

  1. In the result section, it is insufficient to clarify the use of biological replicates and experimental controls. For example, in hypoxia experiments, it is unclear whether non-transfected or non-BN cell lines were consistently used as controls.
  2. The reversible induction of EGFP and endothelial markers in BN cells under hypoxia is intriguing and represents a key strength of the study. However, the underlying mechanism remains insufficiently defined. It is important to clarify whether this phenotype represents true trans-differentiation into endothelial-like cells or a transient, stress-induced upregulation of endothelial-associated genes. To strengthen this interpretation, the authors are encouraged to assess additional canonical endothelial lineage markers, such as VE-cadherin or CD31, and to perform functional assays such as acetylated LDL uptake or tube formation, to demonstrate acquisition of endothelial cell function.
  3. The authors claim that the HB cell lines exhibit comparable sensitivities to conventional chemotherapeutic agents such as cisplatin and doxorubicin; however, this conclusion is not sufficiently supported by the current data presentation. To enhance the rigor of this analysis, statistical comparisons of IC50 values across the different genotypes, for example, using ANOVA with appropriate post-hoc testing should be included. Additionally, providing 95% confidence intervals for the IC50 estimates in Figure 8B would help convey the degree of variability and improve the interpretability of the results.
  4. The extrapolation of murine EC-specific transcript expressions to human cancer datasets, such as TCGA, is ambitious and potentially informative, however, it raises concerns regarding the cellular origin of the observed gene expression patterns. Specifically, it is unclear whether the 1,853 EC-associated genes analyzed were filtered to distinguish tumor-intrinsic expression from that originating in stromal or vascular components. To strengthen the validity of these findings, the authors should clarify whether single-cell RNA-sequencing data or computational deconvolution tools were employed to differentiate tumor cell expression from that of the tumor microenvironment.
  5. Minor comments and corrections: There are a few small issues that should be addressed to improve clarity and readability.
    (1) Typographical errors: Fix minor mistakes such as “noy” → “not” (line 68), “pairwise of triple” → “pairwise or triple” (line 21), “HJ&E” → “H&E” (line 322) and ensure consistent terminology throughout the manuscript, for example: “trans-differentiation” vs. “EC-like properties”.
    (2) Abstract and summary: The Simple Summary is too technical, consider simplifying it for a broader audience. Also, the Abstract should clearly state that EC-like trans-differentiation is reversible and functionally important.
    (3) Figure legends: The figures and charts are not so clear, please ensure all of them are easy to read, with clear legends and axis labels.
    (4) Ethics Statement: Although animal care is mentioned in the Methods, please consider adding a brief ethics statement in the manuscript footer to highlight compliance with ethical standards.
Comments on the Quality of English Language

This manuscript is well written with appropriate scientific language. The narrative is coherent, and the methodology and results are well structured. However, a few typographical and grammatical inconsistencies remain and should be corrected as I commented before. Besides, the Simple Summary and Abstract could be improved to enhance clarity and make the text more accessible to broader readers. It is recommend to have a final round of careful proofreading to ensure the manuscript align with the journal’s standards for publication.

Author Response

1. Comment: “In the result section, it is insufficient to clarify the use of biological replicates and experimental controls. For example, in hypoxia experiments, it is unclear whether non-transfected or non-BN cell lines were consistently used as controls.”

Response. In the revised manuscript’s Materials and Methods section and figure legends, we have now clearly stated the number of biological replicas that were used in each experiment.  We also now mention that all “control” BN and YN cell lines are isologous and transfected with the appropriate “empty” vector.  

  1. Comment: “The reversible induction of EGFP and endothelial markers in BN cells under hypoxia is intriguing and represents a key strength of the study. However, the underlying mechanism remains insufficiently defined. It is important to clarify whether this phenotype represents true trans-differentiation into endothelial-like cells or a transient, stress-induced upregulation of endothelial-associated genes. To strengthen this interpretation, the authors are encouraged to assess additional canonical endothelial lineage markers, such as VE-cadherin or CD31, and to perform functional assays such as acetylated LDL uptake or tube formation, to demonstrate acquisition of endothelial cell function.”

Response: In our original manuscript, Fig. 6G showed the overall changes in expression for 1853 non-redundant, EC-specific genes but did not indicate their individual identities.  This has now been addressed with a new Supplementary File (Supplementary File 2) that names these genes and indicates their changes in expression and q-values. Regarding the Reviewer’s specific question, we quantified the expression for several genes that are considered to be well-accepted, specific and common markers of ECs regardless of source (Fig. 1 and ref. 4 below). In addition to these and other global gene expression changes, the functional robustness of the EC phenotype is shown by the ability of trans-differentiated HB cells to form “tubes” even in the absence of facilitating factors such as specialized/conditioned medium or basement membrane extracts such as Matrigel (Fig. 4G) (1).

Figure 1

Fig. 1. Changes in the expression of standardly accepted EC-specific genes during the course of BN trans-differentiation in response to hypoxia as shown in Fig. 6A of the revised manuscript  (3,4). For the precise fold-changes of all 1853 EC-specific genes among groups, see Supplementary File 2 for a comprehensive listing of all 1853 EC-specific gene changes during the course of BN cell trans-differentiation. UN = unsorted normoxia, UH = unsorted hypoxia; SN = sorted normoxia; SHG = sorted hypoxia.

  1. Comment: “The authors claim that the HB cell lines exhibit comparable sensitivities to conventional chemotherapeutic agents such as cisplatin and doxorubicin; however, this conclusion is not sufficiently supported by the current data presentation. To enhance the rigor of this analysis, statistical comparisons of IC50 values across the different genotypes, for example, using ANOVA with appropriate post-hoc testing should be included. Additionally, providing 95% confidence intervals for the IC50 estimates in Figure 8B would help convey the degree of variability and improve the interpretability of the results.”

Response: Fig. 8B of our revised manuscript (also shown below: Fig. 2) has now been revised to indicate confidence intervals. As we originally concluded, no statistically significant differences in ICâ‚…â‚€ values were observed among the HB cell lines for any of the four drugs examined.

Figure 2

Fig. 2. Chemotherapeutic drug sensitivities of all HB cell lines tested. Each point represents the calculated IC50 for one cell line from the indicated molecular group based on dose-response profiles generated as described in panel A of Figure 8 of the revised manuscript.  Error bars indicate SEM (standard error of the mean).

  1. Comment: “The extrapolation of murine EC-specific transcript expressions to human cancer datasets, such as TCGA, is ambitious and potentially informative, however, it raises concerns regarding the cellular origin of the observed gene expression patterns. Specifically, it is unclear whether the 1,853 EC-associated genes analyzed were filtered to distinguish tumor-intrinsic expression from that originating in stromal or vascular components. To strengthen the validity of these findings, the authors should clarify whether single-cell RNA-sequencing data or computational deconvolution tools were employed to differentiate tumor cell expression from that of the tumor microenvironment.”

Response: The standardized data in TCGA represent gene expression levels that were derived from total tumor samples at a time prior to the development of scRNAseq methodologies and cannot distinguish the cellular origin of EC-specific gene transcripts. Our purpose in showing these results was simply to indicate that, in some tumors, the levels and/or patterns of these transcripts correlate inversely with survival regardless of their origin. This point is now made clearer in the revised manuscript.  These data are also consistent with the findings of others based on the number of actual ECs that are associated with various tumors (2,5).

Minor comments

  1. Comment: Minor comments and corrections: There are a few small issues that should be addressed to improve clarity and readability.
    (1) Typographical errors: Fix minor mistakes such as “noy” → “not” (line 68), “pairwise of triple” → “pairwise or triple” (line 21), “HJ&E” → “H&E” (line 322) and ensure consistent terminology throughout the manuscript, for example: “trans-differentiation” vs. “EC-like properties”.
    (2) Abstract and summary: The Simple Summary is too technical, consider simplifying it for a broader audience. Also, the Abstract should clearly state that EC-like trans-differentiation is reversible and functionally important.
    (3) Figure legends: The figures and charts are not so clear, please ensure all of them are easy to read, with clear legends and axis labels.
    (4) Ethics Statement: Although animal care is mentioned in the Methods, please consider adding a brief ethics statement in the manuscript footer to highlight compliance with ethical standards.

Response: These have all been addressed and/or corrected in the revised manuscript. 

  Specifically, with regard to Point 2 above, we state in the revised Abstract that maintaining BN HB cells under hypoxic conditions for as little as 2 days “reversibly up-regulated the expression of numerous endothelial cell (EC)-specific genes” and that the cells “acquired EC-like properties that benefited tumor growth.”

     With regard to Point 3: Figures provided by the Cancers editorial office were necessarily of low resolution and are hopefully improved in our current resubmission.

References

  1. DeCicco-Skinner KL, Henry GH, Cataisson C, Tabib T, Gwilliam JC, Watson NJ, Bullwinkle EM, Falkenburg L, O'Neill RC, Morin A, Wiest JS. Endothelial cell tube formation assay for the in vitro study of angiogenesis. J Vis Exp. 2014 Sep 1;(91):e51312.

  1. den Uil SH, van den Broek E, Coupé VMH, Vellinga TT, Delis-van Diemen PM, Bril H, Belt EJT, Kranenburg O, Stockmann HBAC, Belien JAM, Meijer GA, Fijneman RJA. Prognostic value of microvessel density in stage II and III colon cancer patients: a retrospective cohort study. BMC Gastroenterol. 2019 Aug 16;19(1):146.

  1. Garlanda C, Dejana E. Heterogeneity of endothelial cells. Specific markers. Arterioscler Thromb Vasc Biol. 1997 Jul;17(7):1193-202.

  1. Goncharov NV, Popova PI, Avdonin PP, Kudryavtsev IV, Serebryakova MK, Korf EA, Avdonin PV. Markers of Endothelial Cells in Normal and Pathological Conditions. Biochem (Mosc) Suppl Ser A Membr Cell Biol. 2020;14(3):167-183.

  1. Hida K, Maishi N, Takeda R, et al. The Roles of Tumor Endothelial Cells in Cancer Metastasis. In: Sergi CM, editor. Metastasis [Internet]. Brisbane (AU): Exon Publications; 2022 May 3. Chapter 10. Available from: https://www.ncbi.nlm.nih.gov/books/NBK580883/

Reviewer 2 Report

Comments and Suggestions for Authors

The manuscript addresses a topic of high scientific and clinical relevance: the derivation and characterization of novel genetically defined murine hepatoblastoma cell lines, with particular focus on their angiogenic potential. The work contributes to an innovative line of research and provides potentially valuable tools for studying tumor progression mechanisms and developing targeted therapeutic approaches. However, several areas of the manuscript require clarification and further elaboration to strengthen its methodological rigor and scientific impact.

Major Revisions

  1. I suggest expanding the introduction to better contextualize the current landscape of preclinical HB models and the main knowledge gaps that this work aims to address.
  2. The section on endothelial transdifferentiation is of interest but requires a more critical discussion of the possible molecular mechanisms involved and comparison with existing literature.
  3. The discussion does not sufficiently address the limitations of the model (e.g., differences between murine and human HB, lack of direct clinical validation). I recommend adding a dedicated section.
  4. Certain methodological aspects for example criteria for cell line selection, standardization of hypoxia conditions, should be described in greater detail to ensure reproducibility.
  5. The conclusion should be strengthened with a forward-looking perspective, emphasizing how these cell lines can be used in translational studies, drug screening, or immunotherapy research.

Minor Revisions

  1. A language revision is recommended to eliminate redundancies and improve fluency.
  2. Figures should be better organized, ideally including schematic summaries linking genetic mutations, cellular phenotype, and tumor behavior.
  3. The reference section should be carefully checked and updated with more recent studies on the topic.
  4. The conclusion would benefit from reinforcement with a few clear take-home messages.
  5. Minor corrections of style, including typos, punctuation, and terminological consistency, are recommended.

Author Response

Major comments

  1. Comment:I suggest expanding the introduction to better contextualize the current landscape of preclinical HB models and the main knowledge gaps that this work aims to address.”

Response: We have now incorporated this suggestion into the Introduction of our revised manuscript (see lines 91-95).

  1. Comment: “The section on endothelial transdifferentiation is of interest but requires a more critical discussion of the possible molecular mechanisms involved and comparison with existing literature.”

Response: This suggestion has now been incorporated into the Discussion of the revised manuscript (see lines 843-852).

  1. Comment: “The discussion does not sufficiently address the limitations of the model (e.g., differences between murine and human HB, lack of direct clinical validation). I recommend adding a dedicated section.”

Response: We thank the Reviewer for this excellent suggestion. We have now included a section at the end of the Discussion that addresses the limitations of our model including the obvious fact that it is not human. At the same time, we discuss how, despite their limitations, these cell lines point the way toward potential approaches for ultimately generating genetically-defined human HB cell lines(see lines 902-916).

  1. Comment: “Certain methodological aspects for example criteria for cell line selection, standardization of hypoxia conditions, should be described in greater detail to ensure reproducibility.”

Response: This has now been done in the Materials and Methods section. Regarding the incubation of our cell lines under hypoxic conditions, we now clearly state that this was achieved by exposing cells to an atmosphere of 1% oxygen + 5% CO2 in a Heracell Vios 160i CO2 incubator (typically for 48 hr).

  1. Comment: “The conclusion should be strengthened with a forward-looking perspective, emphasizing how these cell lines can be used in translational studies, drug screening, or immunotherapy research.”

Response: This has also been added to the end of the Discussion section of the revised manuscript.

Minor comments

  1. A language revision is recommended to eliminate redundancies and improve fluency.
  2. Figures should be better organized, ideally including schematic summaries linking genetic mutations, cellular phenotype, and tumor behavior.
  3. The reference section should be carefully checked and updated with more recent studies on the topic.
  4. The conclusion would benefit from reinforcement with a few clear take-home messages.
  5. Minor corrections of style, including typos, punctuation, and terminological consistency, are recommended.

Response: Together with the other additions/corrections cites above, we have made these recommended revisions.

References

  1. DeCicco-Skinner KL, Henry GH, Cataisson C, Tabib T, Gwilliam JC, Watson NJ, Bullwinkle EM, Falkenburg L, O'Neill RC, Morin A, Wiest JS. Endothelial cell tube formation assay for the in vitro study of angiogenesis. J Vis Exp. 2014 Sep 1;(91):e51312.

  1. den Uil SH, van den Broek E, Coupé VMH, Vellinga TT, Delis-van Diemen PM, Bril H, Belt EJT, Kranenburg O, Stockmann HBAC, Belien JAM, Meijer GA, Fijneman RJA. Prognostic value of microvessel density in stage II and III colon cancer patients: a retrospective cohort study. BMC Gastroenterol. 2019 Aug 16;19(1):146.

  1. Garlanda C, Dejana E. Heterogeneity of endothelial cells. Specific markers. Arterioscler Thromb Vasc Biol. 1997 Jul;17(7):1193-202.

  1. Goncharov NV, Popova PI, Avdonin PP, Kudryavtsev IV, Serebryakova MK, Korf EA, Avdonin PV. Markers of Endothelial Cells in Normal and Pathological Conditions. Biochem (Mosc) Suppl Ser A Membr Cell Biol. 2020;14(3):167-183.

  1. Hida K, Maishi N, Takeda R, et al. The Roles of Tumor Endothelial Cells in Cancer Metastasis. In: Sergi CM, editor. Metastasis [Internet]. Brisbane (AU): Exon Publications; 2022 May 3. Chapter 10. Available from: https://www.ncbi.nlm.nih.gov/books/NBK580883/

Round 2

Reviewer 2 Report

Comments and Suggestions for Authors I have reviewed the revised manuscript. The authors have addressed all comments and requests in a thorough and satisfactory manner. The manuscript is now clear, well-structured, and ready for publication.